# The CXCL10/CXCR3 Pathway Contributes to the Synergy of Thermal Ablation and PD-1 Blockade Therapy against Tumors

**DOI:** 10.3390/cancers15051427

**Published:** 2023-02-23

**Authors:** Wenlu Xiao, Hao Huang, Panpan Zheng, Yingting Liu, Yaping Chen, Junjun Chen, Xiao Zheng, Lujun Chen, Jingting Jiang

**Affiliations:** 1Department of Tumor Biological Treatment, Jiangsu Engineering Research Center for Tumor Immunotherapy, Institute of Cell Therapy, The Third Affiliated Hospital of Soochow University, Changzhou 213003, China; 2State Key Laboratory of Pharmaceutical Biotechnology, Nanjing University, Nanjing 210016, China

**Keywords:** thermal ablation, microwave ablation, CXCL10, tumor microenvironment, cancer immunotherapy

## Abstract

**Simple Summary:**

Thermal ablation has been confirmed to increase the percentage of functional CD8^+^T cells in the tumor microenvironment (TME) and activate tumor-specific T cells via the enhancement of tumor antigen presentation to the immune system. However, its underlying mechanism remains largely unclear. Herein, we demonstrated that the interaction between macrophages and effector CD8^+^T cells through the CXCL10/CXCR3 pathway in the TME contributed essentially to the therapeutic effect of a combined strategy and improved the synergistic effect of this combination therapy of thermal ablation and PD-1 blockade against tumors.

**Abstract:**

As a practical local therapeutic approach to destroy tumor tissue, thermal ablation can activate tumor-specific T cells via enhancing tumor antigen presentation to the immune system. In the present study, we investigated changes in infiltrating immune cells in tumor tissues from the non-radiofrequency ablation (RFA) side by analyzing single-cell RNA sequencing (scRNA-seq) data of tumor-bearing mice compared with control tumors. We showed that ablation treatment could increase the proportion of CD8^+^T cells and the interaction between macrophages and T cells was altered. Another thermal ablation treatment, microwave ablation (MWA), increased the enrichment of signaling pathways for chemotaxis and chemokine response and was associated with the chemokine CXCL10. In addition, the immune checkpoint PD-1 was especially up-regulated in the infiltrating T cells of tumors on the non-ablation side after thermal ablation treatment. Combination therapy of ablation and PD-1 blockade had a synergistic anti-tumor effect. Furthermore, we found that the CXCL10/CXCR3 axis contributed to the therapeutic efficacy of ablation combined with anti-PD-1 therapy, and activation of the CXCL10/CXCR3 signaling pathway might improve the synergistic effect of this combination treatment against solid tumors.

## 1. Introduction

Thermal ablation mainly uses the heat generated by radiofrequencies, microwaves, or lasers to cause coagulative necrosis of local tumors. It has been considered a necessary minimally invasive treatment for certain solid tumors [1,2]. Moreover, thermal ablation is widely accepted as an important form of tumor immunotherapy [3]. After tumor ablation, it can cause local coagulation and necrosis, and tumor antigens are released, stimulating T-cell-mediated anti-tumor immune responses [4]. Our previous study has demonstrated that radiofrequency ablation (RFA) synergizes with PD-1 blockade-based cancer immunotherapy, illustrating the great potential of thermal-ablation-based cancer immunotherapy [5]. Increasing the numbers of tumor-infiltrating T lymphocytes in tumor tissues is an essential step to improve the efficacy of immunotherapy against solid tumors [6,7]. RFA, as well as microwave ablation (MWA), has been confirmed as the most widely used thermal ablation modality in the treatment of solid tumors, including primary liver cancer, colorectal cancer (CRC), liver metastasis, and kidney cancer [8,9,10,11]. In addition, many studies have demonstrated that thermal ablation can affect the microenvironment of solid tumors and promote the infiltration of T lymphocytes into tumor tissues [12,13]. However, the immune response induced by ablation is insufficient to prevent the tumor from recurring. The intensity of the anti-tumor immune response after ablation cannot be maintained or even rapidly weakened, which may be a fundamental reason limiting the efficacy of tumor-ablation treatment [5].

Immune checkpoint blockade (ICB) therapy destroys harmful immunomodulatory molecules and releases pre-existing anti-tumor immune effects by therapeutic antibodies [14,15]. Antibodies against checkpoint molecules have been successfully applied in clinical practice, such as CTLA-4, PD-1, and PD-L1 [16]. Contrarily, ICB-based cancer immunotherapy significantly extends the life span of patients with responsive cancers. In contrast, ICB therapy does not benefit a sizable portion of patients with non-responsive cancers [15,17]. Therefore, it is necessary to explore novel strategies to improve ICB efficacy against solid tumors. Chemokines are cytokines or signal proteins secreted by immune cells, stromal cells, and epithelial cells that can regulate the homing and retention of immune cells in inflamed tissues [18,19]. The CXC chemokine superfamily member CXC-chemokine ligand 10 (CXCL10), also called interferon-γ-inducible protein (IP-10), interacts with the CXC-chemokine receptor 3 (CXCR3) to regulate the immune response, angiogenesis, apoptosis, and proliferation [20,21]. CXCL10 can be up-regulated by IFN-α, IFN-β, IFN-γ, or LPS in various cells, including endothelial cells, fibroblasts, monocytes, and neutrophils. In addition to inducing effector Th1 cells, CXCL10 can also recruit CXCR3^+^CD8^+^T cells to the tumor site and promote these cells to produce Granzyme B, which enhances the anti-tumor effect [22,23].

In the present study, we investigated the role of CXCL10 in thermal-ablation-based tumor immunotherapy by analyzing scRNA-seq data of CD45^+^ immune cells in tumors from the non-thermal-ablation side of Panc02 tumor-bearing mice from a published database [24]. We found that the interaction between macrophages and effector CD8^+^T cells through the CXCL10/CXCR3 pathway in the tumor microenvironment (TME) of the non-ablation zone was required. We then constructed the tumor model by symmetrically injecting the MC38 cells subcutaneously into the bilateral flanks of wild-type and CXCL10 knockout mice according to our previous study. Moreover, we investigated the role of CXCL10 in MWA therapy, ICB therapy, or MWA combined with ICB for liver metastasis of intestinal cancer. Our present study provided valuable insights into the mechanism of thermal-ablation-based tumor immunotherapy and demonstrated that CXCL10 promoted CTLs to migrate into tumor tissues to enhance the synergistic anti-tumor effect of thermal ablation in combination with ICB. Therefore, augmenting the intra-tumoral function of the CXCL10/CXCR3 axis could improve clinical therapeutic efficacy.

## 2. Materials and Methods

### 2.1. Cell Lines and Animals

The murine colon cancer cell line MC38 was obtained from the Shanghai Institutes for Biological Sciences, Chinese Academy of Sciences. MC38 cells were maintained in DMEM (Gibco, Thermo Fisher Scientific, Waltham, MA, USA) supplemented with 100 μg/mL streptomycin, 100 U/mL penicillin, and 10% (*v*/*v*) fetal bovine serum (FBS, Gibco, Thermo Fisher Scientific, Waltham, MA, USA). Cells were cultured for less than 2 weeks before injection into mice. Wild-type C57BL/6 mice and *Cxcl10* knockout mice (*Cxcl10^−/−^*, C57BL/6 background, male, 6–8 weeks old) were obtained from the Model Animal Research Institute of Nanjing University (Jiangsu, China). Animals were housed at the specific pathogen-free (SPF) facility of Cavens Laboratory Animals (Changzhou, Jiangsu, China). All animal experiments were approved by the Ethics Committee of the Third Affiliated Hospital of Soochow University.

### 2.2. Tumor Model, MWA Treatment, and Anti-PD-1 Therapy

Briefly, 2 × 106 MC38 cells were subcutaneously injected into the symmetrical bilateral back of male C57BL/6 and *Cxcl10*^−/−^ mice, and MWA was performed on the right tumor when the tumor volume reached approximately 300 mm^3^. The operation was conducted as reported in our previous study [25]. The ablation needle (Microwave Ablation Antennas, Canyon Medical Inc., Jiangsu Nanjing, China) was inserted percutaneously perpendicular to the center of the tumor with a 1 cm tip. The target temperature was 70 °C, the output power was 10W, and the ablation was performed for 3–5 min to completely ablate the tumor. Then the anti-PD-1 (Clone: J43, BioXcell, Lebanon, NH, USA) therapy was performed on the 2nd day after ablation by intraperitoneally (i.p.) injecting 200 μg antibodies every 3 days four times. Mouse IgG2b (BE0086, BioXcell) was used as a control. Tumor growth was monitored every 2 days after tumor inoculation or MWA treatment. The formula for calculating tumor volume was as follows: Volume = (π × long axis × short axis^2^)/6.

### 2.3. Flow Cytometry

Tumors were collected from mice, cut into pieces smaller than 1 mm^3^, and digested with Liberase TL (0.25 mg/mL, REF 05401020001, Roche) and DNase I (0.33 mg/mL, REF 10104159001, Roche) at 37 °C for 30 min. Subsequently, digestion was terminated by the addition of a serum-containing medium, and the pieces were ground and filtered through a 200-μm filter to obtain a single-cell suspension. Anti-mouse antibodies were used to stain cells, including antibodies against CD45 (Clone 30-F11), Ghost (Cell Signaling Technology, Danvers, MA, USA), CD3 (Clone 17A2), CD4 (Clone GK1.5), CD8 (Clone 53–6.7), Foxp3 (Clone FJK-16S), and PD-1 (Clone RPM1-30). For the measurement of intracellular cytokine levels, the cells were stimulated with a cell-activation cocktail (with brefeldin A, BioLegend, San Diego, CA, USA) at 37 °C for 6 h. After stimulation, the cells were stained with antibodies against surface markers, fixed, and permeabilized according to the manufacturer’s instructions provided by the Invitrogen Fixing/Permeabilization Solution kit. The fixed cells were stained using antibodies against IFN-γ (Clone XMG1.2). Data were acquired using a BD FACS Aria II flow cytometer and were analyzed using FlowJo software.

### 2.4. Isolation of Tumor-Infiltrating CD8^+^T Cells

Briefly, 2 × 10^6^ MC38 cells were subcutaneously injected into the symmetrical bilateral back of male C57BL/6 mice. MWA was performed only on the right flank tumors as mentioned above. After 3 days of MWA treatment, target cells were enriched using CD8α microbeads (Miltenyi) and flow-sorted using FACS Aria II. Total RNA was prepared and subjected to RNA sequencing (Shanghai OE Biotech, Shanghai, China).

### 2.5. Analysis of RNA-Seq Data

Magnetic-bead-enriched and flow-sorted CD8^+^T cells were subjected to mRNA isolation, fragmentation, reverse transcription, and cDNA amplification. These data were analyzed for differential expression by comparing cDNA libraries. Differences between the two groups of samples were assessed using GO term enrichment analysis or gene set enrichment analysis (GSEA) software.

### 2.6. Analysis of scRNA-Seq Data

Sra files were downloaded from the NCBI website. The scRNA-seq data were derived from infiltrating CD45^+^ immune cells established by Fei et al. [24]. Then fastq-dump was used to transfer all sra files into FASTQ files. Next, FASTQ files were processed with cellranger software (Version 4.0.0), which aligned samples to the mm10 genome, followed by filtration and qualification. Finally, cellranger was used to aggregate the data for each sample, and after normalization, the data were merged into one. Seurat was used to analyze the combined data. Doublets were filtered with DoubletFinder (Version 2.0.3), and cells with less than 400 or more than 5500 gene counts, more than 5% mitochondrial contamination, more than 40% ribosome contamination, or more than 25,000 RNA counts were filtered out, leaving 26,084 cells for downstream analysis. Data were normalized and scaled, principal component analysis (PCA) was performed, and neighbors were found using 50 dimensions. PCA was performed on ~3000 genes with the PCA function. The batch effect was removed with harmony (Version 0.1.0). The scaled matrix was subjected to dimensional reduction based on the first 20 PCA components, allowing the cell state to be represented in two dimensions. The main cell clusters were identified with the FindClusters function. Every cell was classified into a known biological cell type using conventional markers.

### 2.7. Differential Gene Expression Analysis

The edgeR package (version 3.28.1) was used to select differentially expressed genes (DEGs) between samples. The raw data obtained from the Seurat object were normalized using TMM (trimmed mean of M-values) with the calcNormFactors function, and the dispersion of gene expression values was estimated with the estimateDisp function. Then, DEGs were selected for the following analysis.

### 2.8. Gene Set Enrichment Analysis

Gene set enrichment analysis was performed using the GSEA software (Version 4.1.0). The gene sets we used were derived via MSigDB gene sets. Difference analyses of pathway activities scored per cell between clusters were performed with wilcox.test.

### 2.9. GO Enrichment Analysis

Differentially expressed genes calculated with the edgeR package were selected for GO analysis with the clusterProfiler R package (Version 3.14.3). The Barplot function was used to visualize the data.

### 2.10. Statistical Analysis

GraphPad Prism V.8 and R software were used in the statistical analyses. The data were presented as mean ± SEM. Two-tailed unpaired Student’s t-tests were used for comparisons between two groups, and the ANOVA was used for multiple comparisons. The log-rank test was used to analyze and calculate the survival of mice. *p* < 0.05 was considered statistically significant.

## 3. Results

### 3.1. scRNA-Seq Identifies Tumor-Infiltrating Immune Cells

We used scRNA-seq data [24] to investigate population changes in infiltrating immune cells in the TME on the distant non-ablation side after thermal ablation treatment. Overall, scRNA-seq data from 26,084 cells were obtained after quality control (Appendix A), including 14,837 cells from the control group and 11,247 cells from the RFA treatment group. We essentially adopted the cell annotations used in the previous study [26], and all cells were assigned to five clusters, including T cells, macrophages, dendritic cells (DCs), neutrophils, and mast cells (Figure 1A,B). Based on the database, we found that thermal ablation treatment increased the proportion of neutrophils and decreased the proportion of DCs and mast cells in the TME on the distant non-ablation side, while the proportion of T cells and macrophages did not change significantly (Figure 1A,C). Therefore, this finding supported the notion that thermal ablation could induce the remodeling of immune cell subsets in the TME.

### 3.2. Ablation Therapy Affects the Interaction of Tumor-Infiltrating Immune Cell Subsets

As a subpopulation of T cells, CD8^+^TILs exert a major anti-tumor effect. We analyzed the RNA-seq data of CD8^+^TILs on the non-ablation-side tumor with flow sorting. Statistical analysis of DEGs was performed for the control and MWA groups. As shown in Figure 2A, CD8^+^TILs in the MWA treatment group were associated with chemotaxis and response to chemokine signatures. Similarly, GO enrichment analysis revealed that cells from the MWA treatment group expressed the transcriptional profile consistent with T cell activation, including leukocyte activation, chemotaxis, and cell adhesion (Figure 2B). Therefore, we speculated that thermal ablation therapy could enhance the anti-tumor effect by promoting lymphocyte infiltration into tumor tissue through chemotaxis.

The scRNA-seq data of the Panc02 tumor-bearing mouse model were analyzed for the strength of the interactions between the immune cell subsets after thermal ablation treatment. We found that macrophages interacted most strongly with other cell type subpopulations, especially with T cells (Figure 3A). To infer the interactions of molecules that mediate the cell–cell interactions in the chemokine system, we calculated the strength of ligand–receptor pairs using the CellphoneDB2 method, which shows the cell population specificity of interaction pairs [27]. Since the response to the chemokine signature was enriched in cells from the ablation treatment group, we collected data on interaction pairs, including chemokine families, and identified multiple interaction pairs that were specific to cell populations. Among them, CXCL10/CXCR3 and CCL2/CCR2 were mainly present between macrophage and T cell interactions (Figure 3B). However, RFA treatment increased the CCL2 expression and decreased the CCR2 expression, which seemed to be somewhat difficult to explain. In contrast, the expressions of CXCL10 and its receptor CXCR3 were both increased in lymphocyte subpopulations after thermal ablation treatment (Figure 3C). Consistent with the existing literature [28], macrophages and neutrophils were the primary sources of CXCL10, and CXCR3 was widely expressed in T cells (Figure 3D). These data indicate that the CXCL10/CXCR3 axis was involved in the anti-tumor immune response induced by thermal ablation.

### 3.3. Ablation-Induced Remodeling of TILs in the TME

We found that the interaction between macrophages and T cells was the closest among tumor-infiltrating immune cells. We isolated the T cells (*n* = 3522) and macrophages (*n* = 10,013) from the other immune cells and performed unbiased cluster classification of cell types using the Seurat package. T cell subsets were divided into seven distinct cell clusters based on the ImmGen database and known cell type markers (Figure 4A,B). Thermal ablation treatment decreased the proportions of ILC, Th, Treg, and cycling CD8^+^T cell clusters compared with the controls and increased the proportion of effector CD8^+^T cells (Figure 4A). Ablation enhanced the activation of CD8^+^T cells, especially cytotoxic CD8^+^T cells, and promoted the accumulation of functional CD8^+^T cells to enhance anti-tumor immunity.

Moreover, we analyzed the macrophage subpopulations in the tumor-infiltrating immune cells based on the cell annotations used in a published study [26]. Tumor-associated-macrophage 1 (TAM1) cells were rich in factors that regulate angiogenesis, such as *Spp1*, *Marco*, and *Vegfa*, while TAM2 cells expressed genes involved in antigen presentation and phagocytosis (*C1qc*, *Trem2*, *Mertk*, and *Cd80*) (Figure 4C). Furthermore, ablation treatment decreased the proportion of TAM2 cells, while the proportion of TAM1 cells was increased (Figure 4C). Moreover, a higher percentage of cells in the ablation group occupied the TAM1 cluster (Figure 4C). Therefore, these were the main interactions between the TAM1 cell subset and T cells after thermal ablation treatment.

To explore the potential mechanism of increased lymphocyte enrichment in TME after thermal ablation treatment, we compared the transcriptomic differences in CD8^+^T cells between control and ablation-treated groups for scRNA-seq data. NicheNet was used to analyze the highly expressed genes in CD8^+^T cells in the ablation treatment group [29]. We noticed that *Cxcl10*, which is highly expressed by TAMs, was predicted to be a potential ligand and drive the CD8^+^T cell effector phenotype in the ablation treatment group (Figure 4D). In addition, among the target genes predicting the phenotype of CD8^+^T cells in the ablation treatment group, *Cxcl10* was one of the prioritized ligands (Figure 4E). It was further proved that CXCR3-induced migration of effector T cells into tumors and CXCL10 production by macrophages were critical for the recruitment of CXCR3-dependent T cells into tumors [30].

### 3.4. CXCL10/CXCR3 Contributes Essentially to Thermal-Ablation-Induced Anti-Tumor Effect

Next, we analyzed the expressions of chemokine-family genes in T cell and macrophage subpopulations both before and after thermal ablation treatment. The expressions of *Cxcl10* and *Cxcr3* were increased after ablation treatment (Figure 5A,B). In addition, *Cxcr3* was essentially expressed in Th, Treg, and naïve CD8 clusters in the control group, whereas the expression of *Cxcr3* was specifically increased in cycling CD8 and effector CD8 clusters in the ablation therapy group (Figure 5A). Therefore, ablation treatment might maintain the expression of *Cxcr3* in effector CD8^+^ T cells. Furthermore, *Cxcl10* was expressed mainly in the TAM1 cluster in the control group and in both TAM1 and TAM2 clusters in the ablation-treated group (Figure 5B). *Cxcr3* is highly expressed in activated T cells and regulates migration behavior and effector function [31,32]. Therefore, we proposed a scientific hypothesis that thermal ablation treatment could induce TAM1 to express CXCL10, leading to the recruitment of CXCR3^+^CD8^+^T cells into the tumor site and enhancement of anti-tumor immunity.

To investigate the function of the CD8^+^T cell subgroup after thermal ablation treatment, we performed GSEA enrichment analysis on the DEGs of CD8^+^T cells in the control and ablation groups. The result revealed that CD8^+^T cells from the ablation group were associated with interferon-alpha/gamma and oxidative phosphorylation, while TNFA signaling via NF-κB, the P53 pathway, and glycolysis were enriched in the control group (Figure 5C). We speculated that thermal ablation treatment induced the expression of CXCL10, stimulated interferon secretion, and recruited CXCR3-expressing CD8^+^T cells to the TME, thus exerting an effector function.

Next, we inoculated MC38 tumor cells bilaterally on the back of C57BL/6 and *Cxcl10*^−/−^ mice and treated one side of the tumor with MWA (Figure 5D). Tumor growth on the non-ablation side was monitored. We observed a slight delay in contralateral tumor growth after MWA treatment (Figure 5E). In contrast, in *Cxcl10*^−/−^ mice, the inhibition of tumor growth by MWA was diminished and the overall survival rate was much lower than that of wild-type mice (Figure 5E,F).

### 3.5. CXCL10 Is Required for Effective Response to PD-1 Blockade Therapy

To explore the role of CXCL10 in the efficacy of anti-PD-1 therapy, we subcutaneously inoculated MC38 cells in wild-type and *Cxcl10^−/−^* mice treated with anti-PD-1 or IgG control antibodies (Figure 6A). The anti-PD-1 treatment could significantly inhibit MC38 tumor growth in wild-type mice, while tumor growth inhibition after anti-PD-1 treatment in *Cxcl10*^−/−^ mice was not as good as in the wild-type mice (Figure 6B). This result demonstrated that CXCL10 played an important role in the anti-PD-1 strategy against MC38 tumors.

We further speculated that the reduced efficacy of anti-PD-1 treatment in *Cxcl10*^−/−^ mice might be related to the reduction in CD8^+^T cells in tumor tissues. The multi-color flow cytometry analysis found that anti-PD-1 treatment increased the ratio of infiltrating CD45^+^ immune cells and CD8^+^T cells in tumors compared with controls from wild-type mice, and there was an increasing trend in the proportion of CD4^+^T cells and Treg cells (Figure 6C). Compared with wild-type mice, the proportion of =CD8^+^T cells in the control and anti-PD-1 treatment groups of *Cxcl10*^−/−^ mice was noticeably reduced, and the proportions of CD4^+^T cells and Treg cells had a decreasing trend (Figure 6C). We next showed that the frequency of IFN-γ^+^CD8^+^TILs cells was similar in wild-type and *Cxcl10*^−/−^ mice in the control group. However, a significant increase in IFN-γ^+^CD8^+^TILs upon anti-PD-1 treatment was found in wild-type mice compared with *Cxcl10*^−/−^ mice (Figure 6D). These data suggest that the chemokine CXCL10 was associated with promoting the CD8^+^T-cell-mediated anti-tumor effects upon anti-PD-1 treatment.

### 3.6. CXCL10 Enhances the Synergistic Anti-Tumor Effect of Thermal Ablation Combined with PD-1 Blockade

Consistent with published reports [24], scRNA-seq data analysis showed increased immune checkpoint PD-1 expression in tumor-infiltrating T cells in distal non-ablated tumors after the ablation treatment (Figure 7A). The MC38 tumor-bearing mouse model also confirmed that PD-1 expression was up-regulated in CD4^+^ and CD8^+^T cells in the non-ablated tumors after MWA treatment (Figure 7B). Combination therapy of ablation and anti-PD-1 significantly enhanced T cell anti-tumor immunity and prolonged the survival of tumor-bearing mice [5]. Next, we studied the role of CXCL10 in the combination therapy of MWA and anti-PD-1 (Figure 7C). We found that the MWA plus anti-PD-1 treatment had a better synergistic anti-tumor effect in MC38-bearing wild-type mice compared with *Cxcl10*^−/−^ mice (Figure 7D).

To better explore the synergistic anti-tumor immune responses of combined ablation and anti-PD-1 therapy, we investigated the T cell response after MWA and MWA plus anti-PD-1. On day 12 after ablation in wild-type mice, the frequency of tumor-infiltrating CD45^+^ and CD8^+^T cells was higher in the combination treatment group than in the MWA group (Figure 7E). Moreover, the proportions of CD45^+^, CD4^+^, and CD8^+^T cells within tumors in *Cxcl10*^−/−^ mice were reduced after combination treatment compared with wild-type mice, especially CD8^+^T cells (Figure 7E). Notably, the proportion of IFN-γ^+^CD8^+^TILs in the combination treatment group was increased compared with the MWA group (Figure 7F). Meanwhile, the proportion of IFN-γ^+^CD8^+^TILs was reduced in *Cxcl10*^−/−^ mice compared with wild-type mice in the MWA group and the combination treatment group (Figure 7F). The combination therapy of MWA and anti-PD-1 in *Cxcl10*^−/−^ mice was less effective than in wild-type mice. These data further support that the anti-PD-1 therapy enhanced MWA-induced anti-tumor effects based on adaptive CD8^+^T cell immune responses, and the chemokine CXCL10 contributed to the synergistic effect of ablation plus anti-PD-1 treatment.

## 4. Discussion

As a widely used strategy of minimally invasive treatment, thermal ablation can destroy tumor tissues, improve the immune recognition of tumor antigens, activate the effective and specific anti-tumor immunity, and especially play an essential role in the initiation of the T-cell-mediated anti-tumor response [33,34]. We have previously demonstrated that in the early-stage CT26-bearing tumor model after RFA, the percentages of the total infiltrating CD8^+^T cells, or the percentages of IFN-γ- or TNF-α-expressing CD8^+^T cells in the tumor tissues, are significantly increased in contrast to the non-ablated groups [5]. These results lead to further considerations about the potential mechanism of how these functional CD8^+^T cells infiltrate into the tumor tissues upon RFA treatment. In our present study, as shown in Figure 2C, based on the scRNA-seq data analysis, *Cxcl10* expressed on myeloid cells and *Cxcr3* expressed on T cells were markedly up-regulated after thermal ablation treatment. Therefore, the CXCL10/CXCR3 axis was indispensable in RFA-triggered anti-tumor immunity.

The chemokine CXCL10 interacts with its receptor CXCR3, and the CXCL10/CXCR3 pathway is well-known to contribute to the migration, differentiation, and activation of many immune cells. It induces an immune response by recruiting immune cells, such as CTLs, natural killer (NK) cells, and macrophages [35,36]. In addition, it can also regulate the polarization of Th1 cells and activate immune cells in response to IFN-γ [35]. Therefore, we assumed that the mechanism of ablation treatment enhancing anti-tumor immune response was regulated by the CXCL10/CXCR3 axis. In addition, we also found that the expression of CXCR3 in cytotoxic CD8^+^T cells was significantly up-regulated after RFA.

It is well known that the involvement of myeloid cells is required to modulate the T-cell-mediated immune response [26]. Based on the results from the intercellular receptor interaction study, we found that the macrophages and T cells might interact through the CXCL10/CXCR3 axis. Further analysis of macrophage subsets showed that the expression of *Cxcl10* in TAM subsets was significantly up-regulated after thermal ablation. Therefore, ablation might promote the recruitment of cytotoxic CD8^+^T cells to infiltrate tumor tissues by up-regulating TAM1 to secrete CXCL10. However, how ablation treatment induces TAM1 to up-regulate CXCL10 in distant non-ablated tumors remains to be further investigated. Moreover, the local inflammation and tumor antigen release induced by ablation can also recruit antigen-presenting cells, including DCs and macrophages, which will prime and activate T cells [37]. However, in the absence of CXCR3 on T cells, DCs could not effectively interact with T cells and then impair the transformation of memory T cells into effector T cells [38]. After ablation treatment, CD8^+^T cells were enriched with interferon-related pathways. Chemokine CXCL10, as an interferon-inducible protein, is most likely involved in the thermal-ablation-induced expression of CXCL10 by TAM1, which stimulates interferon secretion and recruits CXCR3^+^CD8^+^T cells to infiltrate into tumor tissues, thus exerting anti-tumor effects. This is only our speculation at present, and we will follow up with this as a focus for an in-depth study.

Although ICB therapy has made a great breakthrough in the treatment of advanced cancers, there still remains a low response rate in many solid tumors to maintain long-term anti-tumor immunity [39]. Previous studies have shown that PD-1 blockade treatment can enhance the T-cell-mediated anti-tumor response and also increase the levels of IFN-γ and CXCL10 in the TME [40,41]. In our present study, we found that the reduction in tumor progression by anti-PD-1 treatment in MC38 tumor models was dependent on CXCL10. We also found that in the MC38 tumors from *Cxcl10* knockout mice, the activation of CD8^+^T cells was significantly inhibited, in contrast to the wild-type mice upon anti-PD-1 treatment. Our results were consistent with the notion that the anti-tumor immune response elicited by ICB requires the involvement of macrophage-derived CXCL9 and CXCL10 [30].

We have previously confirmed that the tumor-infiltrating T cells expressing PD-1 are significantly increased in the late stage after thermal ablation in contrast to the non-ablated group [5]. This may be an important reason for limiting the inability to maintain the efficacy of thermal ablation treatment. When we carried out the combined treatment of MWA and anti-PD-1, both the numbers and function of the tumor-infiltrating CD8^+^T cells were obviously increased, in contrast to the MWA group, anti-PD-1 group, and control group [5]. A phase I clinical trial has also confirmed that the combined strategy of RFA and anti-CTLA-4 treatments against human hepatocellular carcinoma can present a synergistic anti-tumor effect and an increasing number of CD8^+^TILs [42]. MWA has been verified to induce a robust immune response, but also could up-regulate the expression of immunosuppressive molecules. We have recently reported that the IFN, CCL, and CXCL pathways were significantly enriched in the MWA plus immune checkpoint inhibitor combination therapy compared with the MWA therapy alone [25,43]. The addition of TIGIT blockade to MWA resulted in the up-regulation of CXCL9 and CXCL10 expression in TAMs and their receptor CXCR3 expression in T cells, which restrain tumor growth and enhance anti-tumor immunity [25]. The interactions among the CXCL pathway and IFN pathway were stronger after combination treatment with MWA and LAG3 blockade, and the expression of specific genes in their pathways were also up-regulated [43]. In our current study, we further demonstrated that the chemokine CXCL10 was involved in the synergistic anti-tumor effect induced by the combination therapy of MWA and anti-PD-1. In the combination of MWA and anti-PD-1 treatment, *Cxcl10* knockout mice had fewer CD8^+^TILs and IFN-γ^+^CD8^+^T cells than wild-type mice. Therefore, CXCL10 could regulate the synergistic effect of the combination treatment of thermal ablation with anti-PD-1, and the detailed mechanism merits further investigation.

## 5. Conclusions

Taken together, our current work revealed that thermal ablation treatment increased the proportion of CD8^+^T cells in the TME in distant non-thermally-ablated areas, enhanced the interaction between macrophages and T cells, and increased the expressions of chemokine CXCL10 and its receptor CXCR3. In addition, CXCL10 played an essential role in mediating the anti-tumor effect of anti-PD-1 treatment and also contributed essentially to the synergistic anti-tumor effects of thermal ablation and anti-PD-1 treatment. It could be an important intervention targeting the CXCL10/CXCR3 signaling pathway to improve the synergistic effect of this combined treatment against solid tumors.

## Figures and Tables

**Figure 1 cancers-15-01427-f001:**
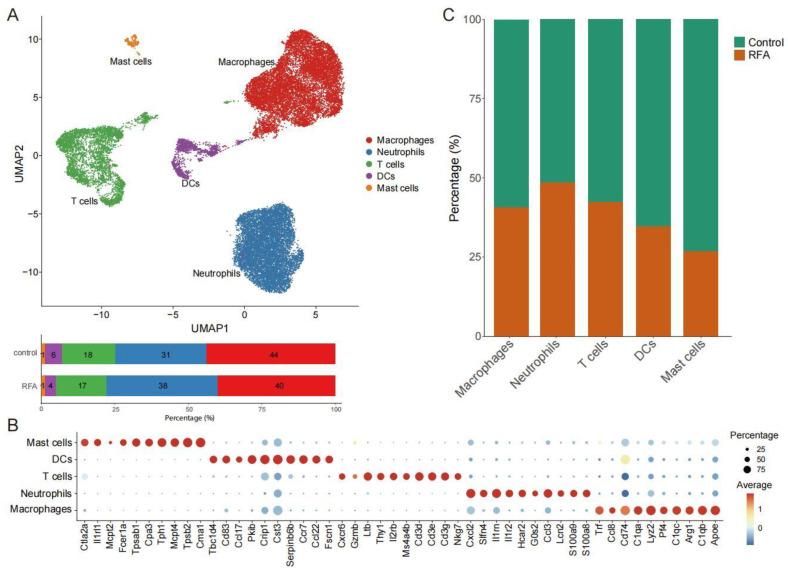
scRNA-seq identifies the changes in tumor-infiltrating immune cells. (**A**). UMAP plot showing CD45^+^ immune cells colored by computationally determined clusters based on scRNA-seq data in the Panc02 tumor-bearing mouse model (**above**). The frequencies of cell composition in the tumor-infiltrating CD45^+^ immune cells with the control group (not subjected to ablation, *n* = 14,837) and ablation group (*n* = 11,247) (**below**). (**B**). Dotplot showing the top 10 marker genes across five different CD45^+^ immune cell subgroups. (**C**). Percentages of cells in the control group and ablation group among the different cell subgroups of CD45^+^ immune cells.

**Figure 2 cancers-15-01427-f002:**
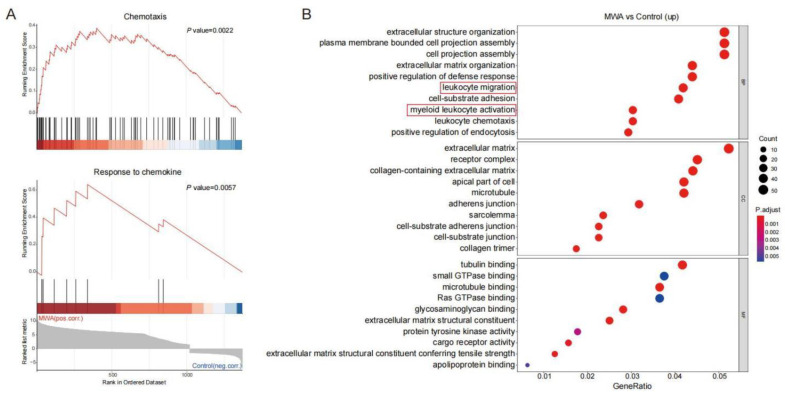
Enrichment analysis of DEGs in CD8^+^TILs after MWA treatment. (**A**). GSEA analysis showing the top enriched chemotaxis and chemokine-mediated signal transduction regulatory pathways in CD8^+^TILs of the MWA treatment group. (**B**). GO enrichment analysis of up-regulated genes in CD8^+^TILs of the MWA treatment group.

**Figure 3 cancers-15-01427-f003:**
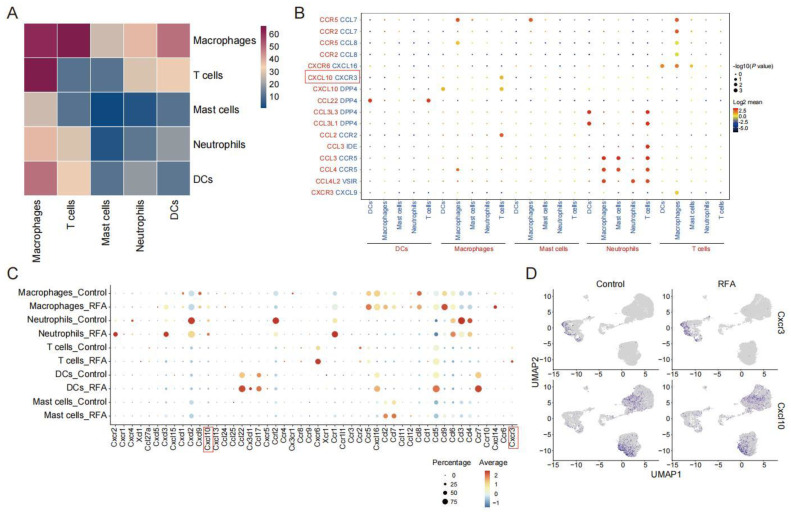
CXCL10-CXCR3 plays a vital role in ablation-induced cell–cell interactions. (**A**). Heatmap showing the cell–cell communication among immune cell populations between subgroups of CD45^+^ immune cells predicted by CellphoneDB2. (**B**). Dotplot showing the expression intensity of selected ligand–receptor pairs among the different cell subgroups of CD45^+^ immune cells. Sizes of dots represent the *p*-value, and colors of dots represent the strength interaction between two subpopulations. (**C**). Dotplot showing the expression levels of chemokines and its receptors across cell types in the control and ablation treatment groups. (**D**). UMAP plot showing expressions of chemokine *Cxcl10* and its receptor *Cxcr3* in two groups.

**Figure 4 cancers-15-01427-f004:**
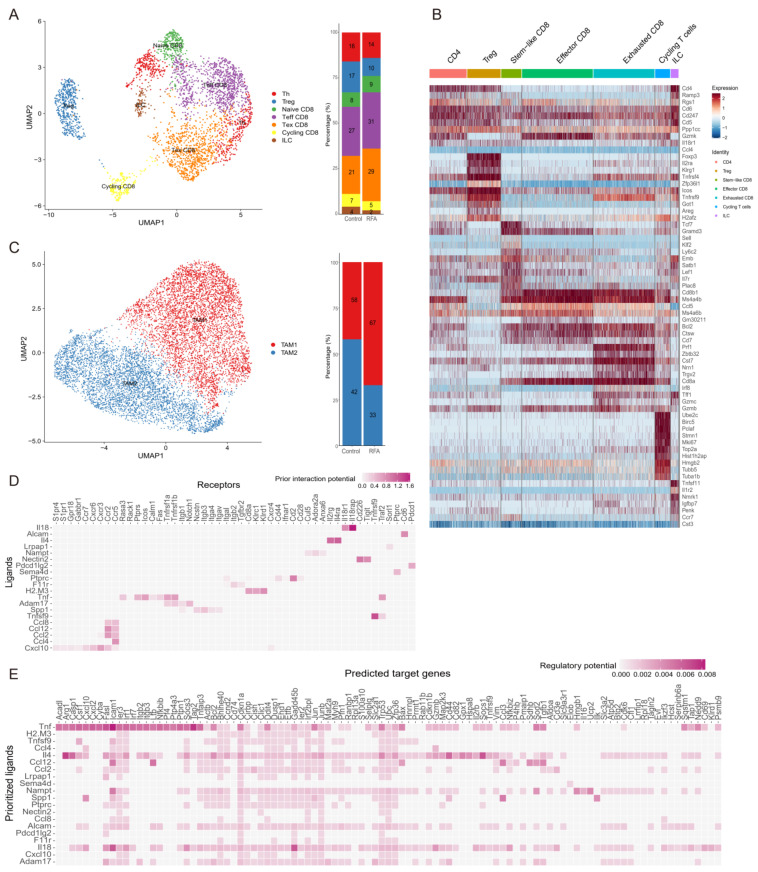
Ablation leads to the remodeling of tumor-infiltrating CD45^+^ immune cell subsets. (**A**). UMAP plot showing sub-clusters of T cells based on scRNA-seq data in the Panc02 tumor-bearing mouse model (**left**). The frequency of cell composition in the T cells of the control group (*n* = 2042) and ablation group (*n* = 1480) (**right**). (**B**). Heatmap displaying marker genes expressed in different subpopulations of TILs. (**C**). UMAP plot showing sub-clusters of macrophages colored by computationally determined clusters (**left**). The frequency of cell composition in the macrophages of the control group (*n* = 5860) and RFA group (*n* = 4153) (**right**). (**D**). Heatmaps showing potential ligands from macrophages interacting with receptors expressed on CD8^+^T cells on the non-ablation side in the ablation group. (**E**). Heatmaps showing that potential ligands from macrophages might influence the gene expression in CD8^+^T cells of the RFA group.

**Figure 5 cancers-15-01427-f005:**
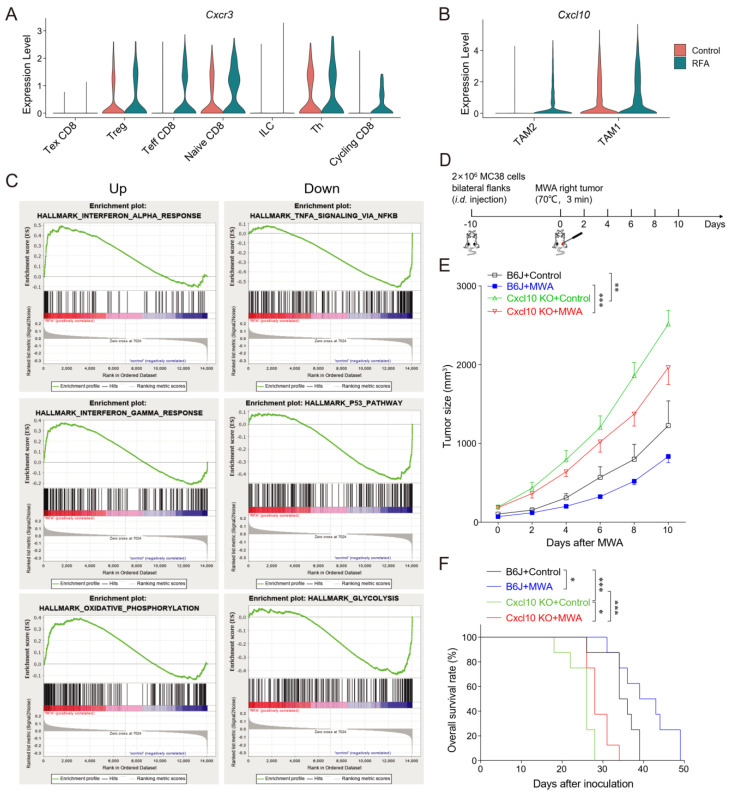
CXCL10 contributes essentially to thermal-ablation-induced anti-tumor effects. (**A**). Normalized expression of the chemokine receptor *Cxcr3* gene in T cell sub-clusters shown by the violin plot. (**B**). Normalized expression of the chemokine *Cxcl10* gene in macrophage sub-clusters shown by the violin plot. (**C**). GSEA analysis showing top-down enriched IFN-α/γ response and oxidative phosphorylation in the MWA group (**left**). GSEA analysis showing top-down enriched TNFA signaling via NF-κB, P53 pathway, and glycolysis in the RFA group (**right**). NES denotes normalized enrichment score. (**D**). Schematic diagram of the protocol for MWA treatment of tumor-bearing mice. C57BL/6 and *Cxcl10*^−/−^ mice were bilaterally inoculated with MC38 cells on the back to construct tumor-bearing mouse models, and one side of the tumor was treated with MWA. (**E**). Tumor burden in C57BL/6 and *Cxcl10*^−/−^ mice treated with MWA (*n* = 6). (**F**). Mouse survival after treatment with MWA in C57BL/6 and *Cxcl10*^−/−^ mice (*n* = 8). Data are presented as the mean ± SEM. * *p* < 0.05, ** *p* < 0.01, and *** *p* < 0.001 according to the one-way ANOVA test and the log-rank test.

**Figure 6 cancers-15-01427-f006:**
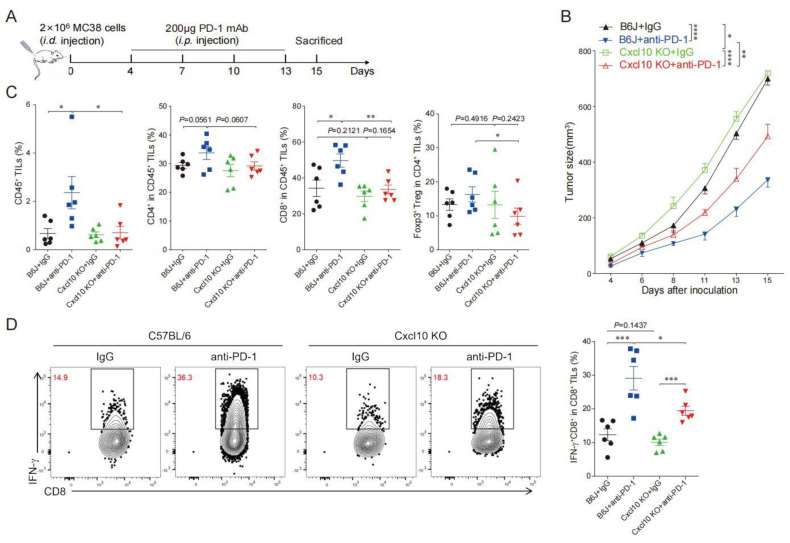
CXCL10 deletion dampens PD-1 blockade efficacy. (**A**). Schematic diagram of the protocol for the anti-PD-1 treatment of tumor-bearing mice. C57BL/6 and *Cxcl10*^−/−^ mice were subcutaneously inoculated with 2 × 10^6^ MC38 tumor cells, then injected i.p. with 200 μg of anti-PD-1 antibody or isotype control antibody on days 4, 7, 10, and 13 after tumor cell inoculation. Tumor growth was monitored until the experimental endpoints. (**B**). Tumor growth in C57BL/6 and *Cxcl10^−/−^* mice treated with control or anti-PD-1 antibody. Six mice were in each group. (**C**). Flow cytometry analysis followed by quantification of CD45^+^ cells, CD4^+^TILs, CD8^+^TILs, and Foxp3^+^Treg within tumors of C57BL/6 and *Cxcl10*^−/−^ mice treated with control or anti-PD-1 antibodies (*n* = 6). (**D**)**.** Representative flow cytometry plots and quantitation of the percentages of IFN-γ expression in CD8^+^TILs (*n* = 6). Data are presented as the mean ± SEM. * *p* < 0.05, ** *p* < 0.01, *** *p* < 0.001, and **** *p* < 0.0001 according to the one-way ANOVA test and the log-rank test.

**Figure 7 cancers-15-01427-f007:**
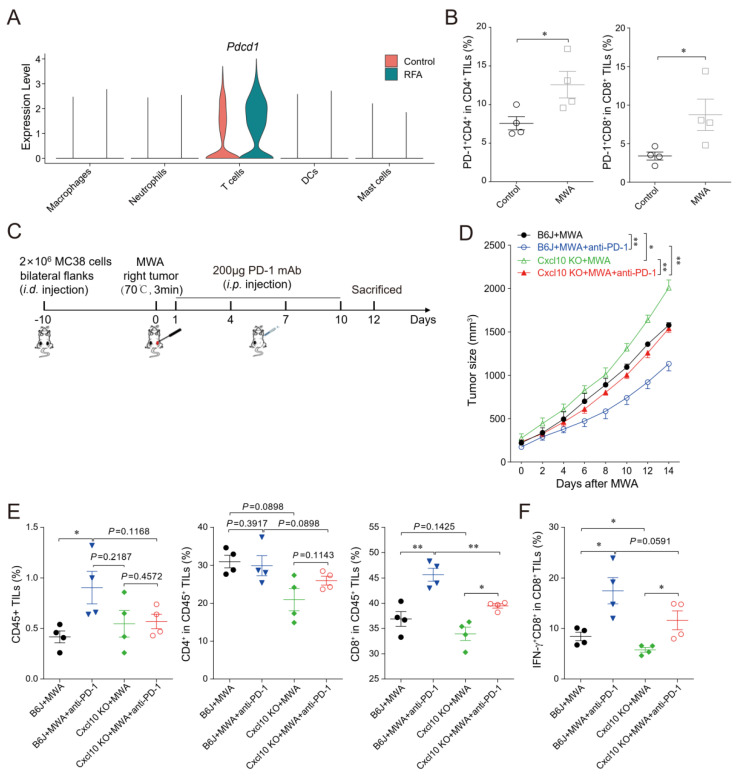
CXCL10 contributes to the synergy of MWA and PD-1 blockade. (**A**). Normalized expression of *Pdcd1* gene in the lymphocyte subsets shown by the violin plot. (**B**). The percentage of PD-1 in CD4^+^TILs and CD8^+^TILs on day 12 of MWA in the MC38 tumor-bearing mouse model dosed with control or MWA treatment. (**C**). Schematic drawing of the study. C57BL/6 and *Cxcl10*^−/−^ mice were bilaterally inoculated with MC38 cells on the back to construct tumor-bearing mouse models, and one side of the tumor was treated with MWA. Mice were injected i.p. with isotype control or anti-PD-1 antibody on day 1 after MWA and then every 3 days four times. (**D**). The size of the tumors on the non-MWA area tumor was recorded every 2 days after MWA. Five mice were in each group. (**E**). Flow cytometric analysis of the percentages of CD45^+^ tumor-infiltrating cells, CD4^+^TILs, and CD8^+^TILs of C57BL/6 and *Cxcl10*^−/−^ mice treated with control or MWA treatment (*n* = 4). (**F**). Representative flow cytometry plots and quantitation of the percentage of IFN-γ expression in CD8^+^TILs (*n* = 4). Data are represented as the mean ± SEM. * *p* < 0.05 and ** *p* < 0.01 according to the one-way ANOVA test and the log-rank test.

## Data Availability

All data generated or analyzed during this study are included in the published article.

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
