# Peer review of "The CXCL10/CXCR3 Pathway Contributes to the Synergy of Thermal Ablation and PD-1 Blockade Therapy against Tumors"

_cancers, 2023, doi:10.3390/cancers15051427_

Round 1

Reviewer 1 Report

To Author, 

It’s my honor to receive the manuscript entitled ‘CXCL10/CXCR3 pathway contributes to the synergy of thermal ablation and PD-1 blockade therapy against tumor’. It explored the role of CXCL10 in thermal ablation-based tumor immunotherapy and provided valuable insights into the mechanism of thermal ablation-based tumor immunotherapy, which is worthy of in-depth study. However, there are still some problems, which must be solved before it is considered for publication. Please see comments below: 

1. The relationship between Cxcl10 and Cxcr3 needs more discussions and please explain the selection reasons.

2. It is suggested that the flow chart of research method can be presented.

3. For the reliability and thoroughness of the results, it would be better if a new experimental group with CXCL10 overexpression mice is added.

4. For the efficiency and accuracy of the result, data process software is expected to upgraded to the latest version.

5. Your paper needs careful editing and particular attention to sentence structure, such as, in paragraph 2, introduction, and paragraph 1, section3.5, “on the other hand” is used to introduce two contrasting facts, but the sentence before and after “on the other hand” has the ascending or coordination relationship.

6. The full name of an abbreviation shall be indicated when the abbreviation appears for the first time, such as, in section 2.3, “Liberase TL” would be marked with the full name “LiberaseThermolysin Low”.

7. Disagreement of English tenses is found in the paper which is expected to be carefully checked, such as, in section 2.4, “we enriched target cells by using CD8α microbeads (Miltenyi) and flow sorting by using FACS Aria II” would be “we enriched target cells by using CD8α microbeads (Miltenyi) and flow sorted by using FACS Aria II” 

8. Please check and double-check the punctuation, spelling and case, for example, in section 2.7, there is a comma at the end of the sentence. In line 5, paragraph 3, section 3.3, the rest of the word “lig-” is missing. In section 3.5, the format and case of “Cxcl10” and “Cxcr3” are not unified.

9. Some sentences contain grammatical mistakes, such as, in section 3.1, “we used scRNA-seq data were derived from infiltrating of CD45+ immune cells.” may be “we used scRNA-seq data derived from infiltrating of CD45+ immune cells.” In section 3.3, “T cell subsets were divided into seven dis- tinct cell clusters based on compared the ImmGendatabase and analyzed known cell type markers” may be “T cell subsets were divided into seven distinct cell clusters based on the ImmGendatabase and known cell type markers.”

10. The number of the result is not in order, “3.4” is missing.

11. The discussion section may need more critical assessments, such as the limitations about you study and what sort of specific future research do you suggest. 

12. It is suggested to enrich the content of conclusion, as it's more of an afterthought. So please highlight important findings and include afterthought of this work in it.

13. Some of the references are in low quality and old. Please update to improve visibility of your manuscript.

Author Response

Dear Editor,

Thank you very much for considering our manuscript “CXCL10/CXCR3 pathway contributes to the synergy of thermal ablation and PD-1 blockade therapy against tumor (Manuscript ID: cancers-2175323)” to be re-considered for publication in Cancers. We have revised our manuscript carefully according to the reviewers’ comments. All the modifications have been highlighted and marked in the revised manuscript. Furthermore, the point-to-point responses have also been listed in this response letter.

We would like to express our great appreciation to you and reviewers for comments on our paper. Look forward to hearing from you.

Thank you and best regards.

Sincerely Yours,

Lujun Chen, MD., Ph.D.

Department of Tumor Biotherapy

The Third Affiliated Hospital of Soochow University

Juqian Road â„–185, Changzhou 213003 P.R.CHINA

Response to reviewers:

It’s my honor to receive the manuscript entitled ‘CXCL10/CXCR3 pathway contributes to the synergy of thermal ablation and PD-1 blockade therapy against tumor’. It explored the role of CXCL10 in thermal ablation-based tumor immunotherapy and provided valuable insights into the mechanism of thermal ablation-based tumor immunotherapy, which is worthy of in-depth study. However, there are still some problems, which must be solved before it is considered for publication.

Response: Thanks for the reviewer’s professional review work on our manuscript. As you’re concerned, there were several problems that need to be addressed. We have made extensive corrections to the previous manuscript based on the reviewers’ suggestions. The detailed corrections are listed below.

Question 1: The relationship between Cxcl10 and Cxcr3 needs more discussions and please explain the selection reasons.

Response: We have made correction according to the reviewer’s comments. CXCR3 is the corresponding receptor for the chemokine CXCL10, which regulates immune response, angiogenesis, apoptosis and proliferation by interacting with its receptor CXCR3. We have further described in the Introduction section of manuscript.

On Page 2. “The CXC chemokine superfamily member CXC-chemokine ligand 10 (CXCL10), also named as interferon-γ-inducible protein (IP-10), interacts with the CXC-chemokine receptor 3 (CXCR3) to regulate immune response, angiogenesis, apoptosis, and proliferation [20, 21].”

Question 2: It is suggested that the flow chart of research method can be presented.

Response: Thank you for the detailed review. In this study, we have made a flowchart presentation in part of the research methodology, including Figure 5D, Figure 6A, and Figure 7C, which can briefly illustrate the elements of relevant research.

Question 3: For the reliability and thoroughness of the results, it would be better if a new experimental group with CXCL10 overexpression mice is added.

Response: Thanks for the reviewer’s comments. Consist with your suggestion, we are constructing CXCL10 overexpressing mice to perform experiments in mouse models. The reliability of the findings has been confirmed by MWA treatment and anti-PD-1 treatment in CXCL10 overexpression mouse models. However, the construction of CXCL10 overexpression mice is a complex process and it will take 6 months to a year to complete.

We also plan to sort CD8+T cells from OT-I mice and overexpress Cxcr3, the receptor for the chemokine Cxcl10, the modified CD8+T cells adoptive transfer into MC38-OVA tumor-bearing C57BL/6 mice to construct adoptive transfer tumor-bearing mouse model. The proportion and changes of OVA+CD8+T cells in the control group and overexpression Cxcr3 group were investigated after MWA treatment and anti-PD-1 treatment in tumor-bearing mice.

These experiments will further confirm the thoroughness of the results, which will also be used and submitted to review in a subsequent research article.

Question 4: For the efficiency and accuracy of the result, data process software is expected to upgraded to the latest version.

Response: Many thanks for the reviewer’s comments. Since the data in this study were processed too early, the data process software might not be the latest version. We updated the GraphPad Prism software used for the statistical analysis of the data in this study to GraphPad Prism version 9.3. After re-analysis of the experimental data, it was found that the trend of the analysis results was consistent with the previous results and did not affect the presentation of the results.

In addition, for the R software used in the RNA-seq data and scRNA-seq data analysis, including the cellranger, DoubletFinder, harmony, GSEA software, clusterProfiler, and the edgeR package, it is not necessary to upgrade all of them to the latest version and re-analyze the data, since the upgraded software applied in our analysis would not greatly affect our results and conclusions. Therefore, we did not make changes to the results in the manuscript, which does not mean that the results are not efficiency and accuracy.

Question 5: Your paper needs careful editing and particular attention to sentence structure, such as, in paragraph 2, introduction, and paragraph 1, section3.5, “on the other hand” is used to introduce two contrasting facts, but the sentence before and after “on the other hand” has the ascending or coordination relationship.

Response: Many thanks for the reviewer’s comments. We have thoroughly checked the grammar of the manuscript. The word “on the other hand” in paragraph 2 of introduction section has been changed to “Contrarily” and the word “on the other hand” in paragraph 1 of section 3.5 has been changed to “Furthermore”. In addition, we examined the entire manuscript and revised any inappropriate structure of sentence.

Question 6: The full name of an abbreviation shall be indicated when the abbreviation appears for the first time, such as, in section 2.3, “Liberase TL” would be marked with the full name “LiberaseThermolysin Low”.

Response: We have made correction according to the reviewer’s comments. We have refined the abbreviations to indicate the full name when the abbreviation appears for the first time, and have listed below the abbreviation appear in the manuscript with their corresponding full names.

On Page 3, section2.3, “LiberaseThermolysin Low (Liberase TL, 0.25 mg/mL, REF 05401020001, Roche) and Deoxyribonuclease I (DNase I, 0.33 mg/mL, REF 10104159001, Roche)”.

Abbreviations: RFA, radiofrequency ablation; MWA, microwave ablation; CRC, colorectal cancer; ICB, immune checkpoint blockade; TME, tumor microenvironment; CXCL10, CXC-chemokine ligand 10; IP-10, interferon-γ-inducible protein; CXCR3, CXC-chemokine receptor 3; PD-1, programmed cell death protein 1; PD-L1, programmed cell death ligand 1; CTLA-4, cytotoxic T lymphocyte-associated antigen-4; GSEA, gene set enrichment analysis; PCA, principal component analysis; DEG, differentially expressed genes; scRNA-seq, single-cell RNA-sequence; DC, dendritic cell; TAM, tumor-associated-macrophage; TIL, tumor infiltrating lymphocyte; Liberase TL, LiberaseThermolysin Low; DNase I, Deoxyribonuclease I.

Question 7: Disagreement of English tenses is found in the paper which is expected to be carefully checked, such as, in section 2.4, “we enriched target cells by using CD8α microbeads (Miltenyi) and flow sorting by using FACS Aria II” would be “we enriched target cells by using CD8α microbeads (Miltenyi) and flow sorted by using FACS Aria II”.

Response: As reviewer suggested, it is really true that we should carefully checked the language tenses in the manuscript. The disagreement of English tenses is found in section 2.4 has been revised to “we enriched target cells were by using CD8α microbeads (Miltenyi) and flow sorted by using FACS Aria II”.

Question 8: Please check and double-check the punctuation, spelling and case, for example, in section 2.7, there is a comma at the end of the sentence. In line 5, paragraph 3, section 3.3, the rest of the word “lig-” is missing. In section 3.5, the format and case of “Cxcl10” and “Cxcr3” are not unified.

Response: We sincerely thank the reviewer for careful reading. We have corrected the punctuation in section 2.7, the word “lig-” into “ligands” in line 5, paragraph 3, section 3.3. We provide an explanation for the inconsistent format and case of Cxcl10 and Cxcr3 in section 3.5. Because the format "Cxcl10" and "Cxcr3" indicates the corresponding gene, and the format "CXCL10" and "CXCR3" indicates the corresponding protein, the different format and case in the manuscript represent different expressions.

Question 9: Some sentences contain grammatical mistakes, such as, in section 3.1, “we used scRNA-seq data were derived from infiltrating of CD45+ immune cells.” may be “we used scRNA-seq data derived from infiltrating of CD45+ immune cells.” In section 3.3, “T cell subsets were divided into seven dis- tinct cell clusters based on compared the ImmGendatabase and analyzed known cell type markers” may be “T cell subsets were divided into seven distinct cell clusters based on the ImmGendatabase and known cell type markers.”

Response: Many thanks for the reviewer’s suggestion. We have revised the grammatical mistakes, including on page 7, in section 3.3, “T cell subsets were divided into seven distinct cell clusters based on the ImmGen database and known cell type markers.” Page 4, in section 3.1 has been revised the “We used scRNA-seq data [24] to investigate changes in infiltrating immune cells in the TME on the distant non-ablation side after thermal ablation treatment.”

Question 10: The number of the result is not in order, “3.4” is missing.

Response: We are very sorry for our negligence of the number of the result was not in order. We have made a revision for this in results section.

Question 11: The discussion section may need more critical assessments, such as the limitations about you study and what sort of specific future research do you suggest. 

Response: Considering the reviewer’s suggestion, we have added more critical assessments to the discussion section.

Question 12: It is suggested to enrich the content of conclusion, as it's more of an afterthought. So please highlight important findings and include afterthought of this work in it.

Response: We have re-written this part according to the reviewer’s suggestion, and enrich the content of conclusion.

Question 13: Some of the references are in low quality and old. Please update to improve visibility of your manuscript.

Response: We realized the inadequacy of the literature review. According to your suggestion, we have updated those references in the revised manuscript.

On Page 15-16, “References 2, 4, 9, 12, 25, 31, and 33”.

Reviewer 2 Report

This is an original manuscript regarding the synergic effect of thermal ablation and PD-1 blockade therapy. The aim of the study and the methods are described exhaustively. English language should be revised.

In “results” should be there are reported only the results of the study, without comments or deductions. Therefore, the following sentence should be moved from this section to the “introduction” or “discussion”

“RFA treatment has been shown to destroy local tumor and induce local and systematic Th1-type immune responses [25] , however, it is still insufficient to consistently prevent cancer progression”

In addition, there are sentences describing the study methods in “results”. Please change the following sentences, describing the details of the experimental design in “methods”:

- “To investigate the limiting factors of thermal ablation-induced tumor immune responses, we used scRNA-seq data were derived from infiltrating of CD45+ immune cells established by Fei et al. [24]”

- “We essentially adopted the cell annotations used in the Zhang et al. study [26] , and all cells were assigned to five clusters, including T cells, macrophages, DCs, neutrophils, and mast cells”

Author Response

Dear Editor,

Thank you very much for considering our manuscript “CXCL10/CXCR3 pathway contributes to the synergy of thermal ablation and PD-1 blockade therapy against tumor (Manuscript ID: cancers-2175323)” to be re-considered for publication in Cancers. We have revised our manuscript carefully according to the reviewers’ comments. All the modifications have been highlighted and marked in the revised manuscript. Furthermore, the point-to-point responses have also been listed in this response letter.

We would like to express our great appreciation to you and reviewers for comments on our paper. Look forward to hearing from you.

Thank you and best regards.

Sincerely Yours,

Lujun Chen, MD., Ph.D.

Department of Tumor Biotherapy

The Third Affiliated Hospital of Soochow University

Juqian Road â„–185, Changzhou 213003 P.R.CHINA

Response to reviewers:

This is an original manuscript regarding the synergic effect of thermal ablation and PD-1 blockade therapy. The aim of the study and the methods are described exhaustively. English language should be revised.

Response: Many thanks for the reviewer’s suggestion. We invited a native English speaker to help polish our article. We hope the revised manuscript could be acceptable for the reviewer. 

Question 1: In “results” should be there are reported only the results of the study, without comments or deductions. Therefore, the following sentence should be moved from this section to the “introduction” or “discussion”

“RFA treatment has been shown to destroy local tumor and induce local and systematic Th1-type immune responses [25], however, it is still insufficient to consistently prevent cancer progression”

Response: Many thanks for the reviewer’s suggestion. Indeed, the sentence was not appropriate at this place, and we have removed it. The relevant content was described in the parts of "introduction" and "discussion".

Question 2: In addition, there are sentences describing the study methods in “results”. Please change the following sentences, describing the details of the experimental design in “methods”:

- “To investigate the limiting factors of thermal ablation-induced tumor immune responses, we used scRNA-seq data were derived from infiltrating of CD45+ immune cells established by Fei et al. [24]”

- “We essentially adopted the cell annotations used in the Zhang et al. study [26], and all cells were assigned to five clusters, including T cells, macrophages, DCs, neutrophils, and mast cells”

Response: Many thanks for the reviewer’s comments. We have agreed with the comments and have modified these sentences carefully.

As shown in Page 4, section2.6," The scRNA-seq data were derived from infiltrating of CD45+ immune cells established by Fei et al.", and paragraph 1of section3.1, " We used scRNA-seq data [24] to investigated changes of infiltrating immune cells in the tumor microenvironment on the distant non-ablation side after thermal ablation treatment".

Reviewer 3 Report

In this study presented by Wenlu Xiao et al. Changes occurred in infiltrating immune cells in a model of tumors developed symmetrically in the same mouse and where only the tumor on one side was treated with MWA and evaluation was performed on the untreated tumor by scRNA-seq. Main emphasis was placed on the CXCL10/CXCR3 recruitment axis. The results are interesting and point to the understanding and development of new therapeutic modalities for the eradication of colon cancer tumors. Even so, there are some aspects that should be reorganized or explained in greater detail in order to improve the quality of the article.

1. It is not clear why the reassessment of the scRNA-seq results from another work was included in this particular article. The relationship that this article intends to demonstrate is carried out in another experimental animal model of another type of cancer. From a personalized medicine point of view, the results obtained in a particular type of tumor should not be extrapolated to another type without a pertinent experimental evaluation.

2. In this sense, what contribution does figure 1 add to this particular work. The difference with the previous reference [24] is not clear.

3. If the intention is to compare the experimental results with those already obtained in ref 24. In my opinion, the results should be reorganized, show the new analyzes first or in supplementary material and then the results of the experimental tests with the MWA model in injected mice. with MC38 cells.

4. MWA treatment modality is poorly described. Please provide a more specific description of how the procedure was carried out in the materials and methods section.

5. If the authors have the possibility of carrying out the RFA treatment as demonstrated in the previous article. It would be a good complement to compare the two therapeutic modalities of thermal ablation (RFA and MWA) in the same experimental model, since the responses are not necessarily the same for these treatment thermal ablative modalities. Please read relavant references: 10.1245/s10434-014-3817-0, 10.4254/wjh.v7.i8.1054.

Author Response

Dear Editor,

Thank you very much for considering our manuscript “CXCL10/CXCR3 pathway contributes to the synergy of thermal ablation and PD-1 blockade therapy against tumor (Manuscript ID: cancers-2175323)” to be re-considered for publication in Cancers. We have revised our manuscript carefully according to the reviewers’ comments. All the modifications have been highlighted and marked in the revised manuscript. Furthermore, the point-to-point responses have also been listed in this response letter.

We would like to express our great appreciation to you and reviewers for comments on our paper. Look forward to hearing from you.

Thank you and best regards.

Sincerely Yours,

Lujun Chen, MD., Ph.D.

Department of Tumor Biotherapy

The Third Affiliated Hospital of Soochow University

Juqian Road â„–185, Changzhou 213003 P.R.CHINA

Response to reviewers:

In this study presented by Wenlu Xiao et al. Changes occurred in infiltrating immune cells in a model of tumors developed symmetrically in the same mouse and where only the tumor on one side was treated with MWA and evaluation was performed on the untreated tumor by scRNA-seq. Main emphasis was placed on the CXCL10/CXCR3 recruitment axis. The results are interesting and point to the understanding and development of new therapeutic modalities for the eradication of colon cancer tumors. Even so, there are some aspects that should be reorganized or explained in greater detail in order to improve the quality of the article.

Response: Many thanks for the reviewer’s comments. We have re-organized and interpreted the manuscript based on the reviewer’s comments in the revised manuscript.

Question 1: It is not clear why the reassessment of the scRNA-seq results from another work was included in this particular article. The relationship that this article intends to demonstrate is carried out in another experimental animal model of another type of cancer. From a personalized medicine point of view, the results obtained in a particular type of tumor should not be extrapolated to another type without a pertinent experimental evaluation.

Response: Many thanks for the reviewer’s comments. We found only one case of scRNA-seq data from Fei et al. established on tumor-infiltrating CD45+ immune cells in the Panc02 tumor-bearing mouse treated with RFA [24]. Therefore, we downloaded this data for analysis to explore changes in the infiltrating immune cells in TME. However, we recently constructed MWA-treated MC38 tumor-bearing mouse model and performed scRNA-seq for CD45+ cells from the non-ablation side of the tumor (Front Immunol 2022, 13,832230. PMID: 35320940; J Transl Med 2022, 20,433. PMID: 36180876). These results are consistent with those obtained from the analysis of the data cited in reference [24]. But this data was not presented in this manuscript.

Question 2: In this sense, what contribution does figure 1 add to this particular work. The difference with the previous reference [24] is not clear.

Response: We are very sorry that the explanation of Figure 1made some incomprehensible understandings. We used scRNA-seq data from the previous reference [24] to reanalyzed the expression of tumor infiltrating CD45+ immune cells before and after ablation treatment. The marker genes defining cell subpopulations slightly differ from those in this manuscript (Cell 2020, 181,442-459 e429. PMID: 32302573). The cell populations defined in the previous reference [24] was not according to the biological functions. We re-defined them and analyzed changes between control and ablation treatment. Based on it, we determined the differences in chemokine CXCL10 expression after ablation treatment and performed subsequent experimental studies.

Question 3: If the intention is to compare the experimental results with those already obtained in ref 24. In my opinion, the results should be reorganized, show the new analyzes first or in supplementary material and then the results of the experimental tests with the MWA model in injected mice. with MC38 cells.

Response: Thanks for the reviewers’ comments. The reason we downloaded and analyzed the scRNA-seq data was aim to confirm the relationship between thermal ablation and CXCL10. The scRNA-seq data in this reference are consistent with the research direction of thermal ablation-induced immune effects that our group has been studying. Our team's previous published articles (Front Immunol 2022, 13,832230. PMID: 35320940; J Transl Med 2022, 20,433. PMID: 36180876) performed scRNA-seq of tumor-infiltrating lymphocytes from the untreated side after MWA treatment of MC38-bearing mice, and revealed that the conclusions were consistent with those obtained in the previous reference we selected [24].

Question 4: MWA treatment modality is poorly described. Please provide a more specific description of how the procedure was carried out in the materials and methods section.

Response: Thanks for the reviewer’s suggestion. We have added reference about MWA treatment modality. MWA treatment in this study was consistent with that in two published papers by our team (Front Immunol 2022, 13,832230. PMID: 35320940; J Transl Med 2022, 20,433. PMID: 36180876). Additionally, we added greater detail about MWA treatment modality into methods section in the manuscript.

Question 5: If the authors have the possibility of carrying out the RFA treatment as demonstrated in the previous article. It would be a good complement to compare the two therapeutic modalities of thermal ablation (RFA and MWA) in the same experimental model, since the responses are not necessarily the same for these treatment thermal ablation modalities. Please read relevant references: 10.1245/s10434-014-3817-0, 10.4254/wjh.v7.i8.1054.

Response: Thanks for the reviewer’s comments. The purpose of this study was aim to research the role of thermal ablation therapy in tumor immunology. The most frequently methods of thermal ablation used in clinical were RFA and MWA. They have few differences in operational approach, but they lead to similar immune effects in tumor treatment. In addition, three published articles by our team (Clin Cancer Res 2016, 22,1173-1184. PMID: 26933175; Front Immunol 2022, 13,832230. PMID: 35320940; J Transl Med 2022, 20,433. PMID: 36180876) have demonstrated that RFA and MWA have similar therapeutic responses in tumor immunotherapy. Moreover, we have investigated the immune effects induced by thermal ablation therapy and immune checkpoint blockade therapy.

Round 2

Reviewer 3 Report

my questions have been answered to my satisfaction and the changes made have improved the reading of the article. I have no more comments.